# The Risk of Gastrointestinal Bleeding between Non-Vitamin K Antagonist Oral Anticoagulants and Vitamin K Antagonists in the Asian Atrial Fibrillation Patients: A Meta-Analysis

**DOI:** 10.3390/ijerph18010137

**Published:** 2020-12-27

**Authors:** Kuang-Tsu Yang, Wei-Chih Sun, Tzung-Jiun Tsai, Feng-Woei Tsay, Wen-Chi Chen, Jin-Shiung Cheng

**Affiliations:** 1Division of Gastroenterology and Hepatology, Department of Internal Medicine, Kaohsiung Veterans General Hospital, Kaohsiung 81362, Taiwan; ktyang1104@gmail.com (K.-T.Y.); wcsun@vghks.gov.tw (W.-C.S.); tjtsai@vghks.gov.tw (T.-J.T.); fwchaie@vghks.gov.tw (F.-W.T.); jcheng@vghks.gov.tw (J.-S.C.); 2School of Medicine, National Yang-Ming University, Taipei 11221, Taiwan; 3Department of Chemistry, College of Science, National Kaohsiung Normal University, Kaohsiung 80201, Taiwan

**Keywords:** gastrointestinal bleeding, Asian, atrial fibrillation, Non-vitamin K antagonist oral anticoagulants, vitamin K antagonists

## Abstract

*Background:* Non-vitamin K antagonist oral anticoagulants (NOACs) are more commonly used to prevent atrial fibrillation (AF) patients from thromboembolic events than vitamin K antagonists (VKAs). However, the gastrointestinal bleeding (GIB) risk in the Asian AF patients associated with NOACs in comparison with VKAs remained unaddressed. *Materials and Methods:* A systematic search of studies on NOACs and VKAs in the Asian AF patients was conducted in PubMed, Cochrane Library, and ClinicalTrials.gov. The primary outcome was the hazard ratio (HR) of any GIB associated with NOACs versus VKAs. The secondary outcome was the GIB risks in different kinds of NOACs compared with VKAs. *Results:* This meta-analysis included two randomized controlled trials (RCTs) and four retrospective studies, comprising at least 200,000 patients in total. A significantly lower HR of GIB risks was found in all kinds of NOACs than VKAs in the Asian AF patients (HR: 0.633; 95% confidence interval: 0.535–0.748; *p* < 0.001). Additionally, the GIB risks of different NOACs were apixaban (HR: 0.392), edoxaban (HR: 0.603), dabigatran (HR: 0.685), and rivaroxaban (HR: 0.794), respectively. *Conclusions:* NOACs significantly reduced the risk of GIB in the Asian AF patients compared with VKAs. In the four NOACs compared with VKAs, apixaban probably had a trend of the least GIB risk. We need further head-to-head studies of different NOACs to confirm which NOAC is the most suitable for Asian AF patients and to know the optimal dosage regimen of different NOACs.

## 1. Introduction

The overall atrial fibrillation (AF) prevalence is about 1% worldwide, and nearly 10% in populations older than 80 years old [1]. Stroke prevention in the AF patients is an important issue because patients with AF have an approximately five times higher risk of stroke than those without AF [2,3]. The resulting mortality and bed-ridden status bring plenty of problems in terms of medical expenditure and long-term care [4].

Non-vitamin K antagonist oral anticoagulants (NOACs) involve dabigatran, which inhibits thrombin, and rivaroxaban, apixaban, and edoxaban, which inhibit factor Xa. NOACs have some advantages, such as minor drug-food or drug-drug interactions and no need for laboratory monitoring. Besides, Asian AF patients under vitamin K antagonists (VKAs, warfarin) use easily encountered bleeding events and would seldom reach an optimal international normalized ratio control when taking VKAs.

Moreover, Asian AF patients have more risks of stroke/systemic embolism, ischemic stroke, and hemorrhagic stroke than non-Asians under VKAs use [5]. Therefore, NOAC use is recommended in non-Asia areas [6,7,8]. However, the gastrointestinal safety remains an essential concern of NOAC use.

Gastrointestinal bleeding (GIB) resulting from NOACs is potentially life-threatening [9,10]. However, the current non-Asia meta-analyses are controversial. According to the meta-analysis performed by Holster et al., a 1.6- and 1.5-fold increased GIB risk among dabigatran and rivaroxaban users was reported, respectively [11]. A subsequent meta-analysis by Rong et al. discovered no increased GIB risk associated with NOACs [12]. On the other hand, a meta-analysis by Wang et al. [13] mentioned that with VKAs use, Asian patients had similar GIB to non-Asian patients, and the risk of GIB with standard-dose NOACs was higher than that with warfarin. However, such the observation might be modified by races. The hypothesis is that the concomitant antiplatelet therapy and the use of proton pump inhibitors might be independent factors to influence the GIB risk. As time goes on, further studies including the GIB risk of the Asian AF patients taking different NOACs (rivaroxaban, dabigatran, apixaban, and edoxaban) compared with VKAs are being published. Therefore, this study aimed to apply the up-to-date trials and real-world studies to investigate the GIB risk in the Asian AF population using NOACs and VKAs. The GI safety of different NOACs versus VKAs was also evaluated.

## 2. Materials and Methods

This meta-analysis was performed by following the guidance of the Preferred Reporting Items for Systematic Reviews and Meta-Analyses (PRISMA) (Appendix A).

### 2.1. Search Strategy and Inclusion Criteria

PubMed, Cochrane Library, and ClinicalTrials.gov were searched from inception to December 2019 without publication date restriction for the studies which are relevant to GIB between NOACs and VKAs in the Asian AF patients. The bibliographies of the included trials and related review articles were manually reviewed for relevant references. Any duplicated records, titles not compatible with our population/intervention/control/outcome (PICO), case report, or abstract were excluded. We ruled out studies without GIB data or appropriate statistical methods, and removed the literature probably extracted from the identical database, the same population, non-Asians, or not fully Asians. We investigated studies with a GIB risk evaluation and employing the Asian AF patients receiving different kinds of NOACs or VKAs. The search strategy (File S2) comprised the following keywords variably combined with rivaroxaban, Xarelto, dabigatran, Pradaxa, apixaban, Eliquis, edoxaban, Lixiana, novel oral anticoagulant, new oral anticoagulant, direct oral anticoagulant, NOAC, DOAC, novel, new, oral, anticoagulant, coagulant, oral anticoagulant, OAC, antithrombin, thrombin, factor Xa inhibitor, Xa inhibitor, factor IIa inhibitor, IIa inhibitor, Non-vitamin K antagonist, gastrointestinal bleeding, GIB, GI bleeding, gastrointestinal, bleeding, gastrointestinal hemorrhage, GI hemorrhage, hemorrhage, warfarin, vitamin K antagonist, VKA, atrial fibrillation, AF, Afib, Asia, Asian, Taiwan, Taiwanese, China, Chinese, Abkhazian, Iran, Palestine, Afghanistan, Iraq, Akrotiri, Dhekelia, Israel, Philippines, Japan, Japanese, Qatar, Armenia, Jordan, Azerbaijan, Kazakh, Bahrain, North Korea, Russia, Bangladesh, Korea, Korean, Saudi Arabia, Bhutan, Kuwait, British Indian Ocean Territory, Kyrgyz, Singapore, Sri Lanka, Brunei, Laos, Syria, Cambodia, Lebanon, Tajik, Macao, Thailand, Christmas Island, Malaysia, East Timor, Cocos, Maldives, Turkey, Turkmenistan, Cyprus, Mongolia, Aliani, Arab, Egypt, Myanmar, Uzbek, Georgia, Nepal, Vietnam, Hong Kong, Oman, Yemen, India, Indian, Pakistan, and Indonesia. Randomized controlled trials (RCTs) and comparative retrospective studies were included. All retrieved studies were required to comprise at least two treatment arms, one of which was NOACs and the other was warfarin. The target population was Asian patients who had AF. Studies that explored the adverse events of NOACS or VKAs and the detailed sites of GIB were beyond the scope of the present meta-analysis.

### 2.2. Data Extraction and Quality Assessment

Two reviewers examined all the retrieved articles and extracted data by using a predetermined form. We recorded the first author, published year, type of interventions, study design, number of patients, average age, data source, country, outcome, GIB hazard ratio (HR), and 95% confidence interval (CI). The methodological quality of enrolled studies was evaluated by two independent reviewers. We used the risk of bias tools (ROB 2.0) for the RCTs [14]. We applied the risk of bias in non-randomized studies of interventions (ROBINS-I) for the retrospective studies [15].

ROB 2.0 evaluates the methodology of RCTs according to five domains, indicating low risk, some concerns, and high risk: bias arising from the randomization process, bias due to deviations from intended interventions, bias due to missing outcome data, bias in measurement of the outcome, and bias in selection of the reported result. ROB 2.0 excluded “other bias” from the previous version.

The ROBINS-I contains three subgroups, including pre-intervention, at-intervention, and post-intervention. Pre-intervention emphasizes bias due to confounding and bias in selection of participants into the study. At-intervention highlights bias in classification of interventions. Post-intervention underlines bias due to deviations from intended interventions, bias due to missing data, bias in measurement of outcomes, and bias in selection of the reported result. The discrepancies between the reviewers were discussed under the supervision of the other authors.

### 2.3. Data Synthesis and Analysis

Before pooling RCTs and retrospective studies, we performed meta-regression analysis to assess the potential difference of reported outcomes between RCTs and retrospective studies. We pooled RCTs and retrospective studies and then separated them into RCTs and retrospective studies subgroups. A random-effects model was employed to pool individual HR and 95% CI of any GIB of NOACs or VKAs users as the primary outcome. The data was extracted from the visual analog scales evaluated. HR less than 1 indicated NOACs to be a favorable treatment option. Different kinds of NOACs compared with VKAs causing GIB comprised the secondary outcome. The Forrest plot was applied to measure the primary and secondary outcomes. All analyses were performed using Comprehensive Meta-Analysis (CMA) software, version 3 (Biostat, Englewood, NJ, USA). I-squared tests were used to determine the between-trial heterogeneity. Funnel plots and Egger’s test were used to examine the potential publication bias [16]. A *p*-value less than 0.05 defined statistical significance, except for the determination of the publication bias, which employed *p* less than 0.10.

### 2.4. GRADE System, Meta-Regressions, and Sensitivity Analyses

The GRADE system was used to grade the quality of evidence [17]. The GRADE system judged evidence of having a lower quality if there were study limitations, inconsistency, indirectness, imprecision, or publication bias. Large effect, dose-response, or plausible confounders were factors that caused higher quality. We made meta-regressions to examine the important and common covariates which might influence the outcomes. We also performed sensitivity analyses by excluding one study at a time and calculating the pooled HRs.

## 3. Results

### 3.1. Study Search and Research Evaluation

We retrieved 1213 records identified through database searching and no additional records identified through other sources. Eighty-four duplicated records were removed. One thousand and twenty-eight incompatible titles, case reports, or abstracts were excluded. One hundred and one full-text articles assessed for eligibility were included (Figure 1). Then sixty-four articles without GIB data or using inappropriate statistical methods were ruled out. Fifteen articles probably extracted from the identical database or the same population were not applied [18,19,20,21,22,23,24,25,26,27,28,29,30,31,32]. Sixteen articles investigating non-Asians or not fully Asians were not enrolled [33,34,35,36,37,38,39,40,41,42,43,44,45,46,47,48].

Finally, six articles were included for meticulous evaluation after eliminating the references violating the inclusion criteria. First author, published year, type of interventions, study design, number of patients, average age, data source, country, and outcome are listed in Table 1. This meta-analysis included two RCTs [49,50], which were appraised by ROB 2.0 (Figure 2a). The other four retrospective studies were enrolled [51,52,53,54] and were evaluated by ROBINS-I (Figure 2b).

### 3.2. Risk of Bias in Enrolled Articles

The risk of bias outcome was based on the Cochrane guidelines [14,15] and is summarized in Figure 2a,b.

The enrolled articles were mainly retrospective studies, belonging to the non-randomized group. They had moderate to serious risk of bias due to confounding. They had serious performance in bias in selection of participants into the study. There was a low risk of bias in the classification of interventions and bias in missing data. Most studies were appraised as having a serious risk of bias due to deviations from intended interventions. All retrospective studies were evaluated as serious risk of bias in measurement of outcomes. Finally, all retrospective studies were appraised as moderate risk of bias in the selection of the reported result.

The two RCTs had low risk in bias arising from the randomization process, bias due to deviations from intended interventions, bias due to missing outcome data, bias in measurement of the outcome, and bias in the selection of the reported result.

### 3.3. Gastrointestinal Bleeding Comparison and Possible Moderators

After adjusting for the study design with meta-regression, the heterogeneity variance was not decreased (pooled: 0.0393; adjusted: 0.0401), which hinted that the mean treatment effect was not different between RCTs and retrospective studies. The estimated difference between these two designs was also not statistically significant (odds ratio (OR) = 1.43; 95% CI: 0.8033 to 2.5459; *p*-value = 0.2241). As shown in Table 2, the outcome after pooling four retrospective studies and two RCTs showed a significantly lower GIB risk of NOACs than VKAs in the Asian AF patients (HR: 0.633; 95% CI, 0.535 to 0.748; *p* < 0.001; I^2^: 61.6%) (Figure 3a). The retrospective subgroup also showed a significantly lower GIB risk of NOACs than VKAs (HR: 0.610; 95% CI, 0.509 to 0.730; *p* < 0.001; I^2^: 68.9%). However, the RCT subgroup revealed a trend toward less GIB risk for NOAC users but did not show statistical significance (HR: 0.864; 95% CI, 0.529 to 1.409; *p* = 0.557; I^2^: 0%) (Figure 3b).

The further analysis showed a ranking from lower to higher risk of NOACs compared with VKAs was apixaban (HR: 0.392; 95% CI, 0.173 to 0.890; *p* = 0.025; I^2^: 82.0%) (Figure 4a), edoxaban (HR: 0.603; 95% CI, 0.434 to 0.839; *p* = 0.003; I^2^: 35.0%) (Figure 4b), dabigatran (HR: 0.685; 95% CI, 0.500 to 0.938; *p* = 0.018; I^2^: 35.4%) (Figure 4c), and rivaroxaban (HR: 0.794; 95% CI, 0.697 to 0.904; *p* = 0.001; I^2^: 0%) (Figure 4d).

An Egger’s test revealed no significant publication bias (*p*: 0.3584). There was neither no publication bias in the retrospective studies, RCTs, edoxaban, rivaroxaban, and dabigatran subgroups. Due to rare articles for apixaban, the publication bias could not be evaluated. The funnel plot showed standard error and log hazard ratio for overall NOACs (Figure 5).

### 3.4. GRADE for Overall

As shown in Table 2, there was a significantly lower hazard ratio for GIB of NOACs than VKAs in the Asian AF patients with moderate heterogeneity (I^2^: 61.6%). Because the overall meta-analysis included a large proportion of retrospective studies, the study limitations downgraded the quality of evidence to low. The overall risk of bias was serious. Imprecision and indirectness did not exist. The publication bias was not likely according to the Egger’s test showing a 2-tailed *p*-value 0.3584, which was >0.1. The HR was more than 0.5, so the large effect was not prominent. Because NOAC dosage regimen had a response in clinical conditions such as treatment efficacy and bleeding association, quality due to dose-response was upgraded. In retrospective studies, there were no obvious plausible confounders. The GRADE system showed very low certainty, which indicated that the confidence in the effect estimate was limited. The true effect might be substantially different from the estimate of the effect. However, there were few studies about the post-marketing evaluation of NOACs in Asia.

### 3.5. GRADE for RCT Subgroup

In RCTs, study limitation was mild due to some concerns in ROB 2.0 appraisal. Heterogeneity did not exist. Imprecision and indirectness were excluded. Publication bias was not evident, either. The HR was more than 0.5, so there was no large effect. Dose-response resulted in an upgrade for the quality of the evidence. There might be some plausible confounders in RCTs due to the short follow up duration for the enrolled patients, which could increase the HR and cause an upgrade for the quality of the evidence. We evaluated high certainty for the RCT group.

### 3.6. GRADE for Retrospective Studies Subgroup

Almost like the pooled outcome, there was a serious risk of bias which downgraded. Inconsistency (I^2^: 68.9%) existed. We found no obvious publication bias, indirectness, or imprecision. Large effect and plausible confounders were not present. Dose-response resulted in an upgrade for the quality of the evidence. The retrospective studies subgroup acquired very low certainty.

### 3.7. GRADE for the Subgroups of Different NOACs

We noticed that from the ranking of the HR of each NOAC, apixaban was known as the best for the GI safety profile. Its large effect (HR: 0.39) caused an upgrade for the quality of evidence, but publication bias was suspected due to the very low HR outcome and few GIB studies for apixaban. Study limitations existed in the four subgroups, whereas the heterogeneity was prominent only in the apixaban subgroup. Dose-response brought about an upgrade in all NOACs.

### 3.8. Meta-Regressions and Sensitivity Analyses

We examined several important and common covariates concerning the study outcomes. As shown in Table 3, the 2-sided *p*-value was >0.05 in age, female ratio, and publication year, which indicated that no covariates showed statistically significant associations in the meta-regression analyses. The meta-regressions of the log hazard ratio on age, female ratio, and publication year were shown in Appendix A. Sensitivity analyses revealed the corresponding results did not change in the direction substantially. For example, when we excluded the study by Lee [54] (the study which carried the most weight) from our analysis, the HR remained statistically significant (HR: 0.572; 95% CI, 0.394 to 0.831; *p* = 0.003), and the Egger’s test showed 2-tailed *p*-Value 0.8930 (no obvious publication bias). This analysis verified the consistency of the lower GIB risk of NOACs than of VKAs.

## 4. Discussion

This study was the first meta-analysis investigating the GIB risk associated with NOACs in Asian AF patients. It highlighted the real situations of GIB resulting from NOACs use in Asia. Our study recruited more than 200,000 patients. We used the rigorous article appraisal tools such as ROB 2.0 for RCTs and ROBINS-I for retrospective studies.

According to this study, overall, NOACs cause less risk of GIB than conventional VKAs. Among them, apixaban seemed to bring about the lowest risk of GIB compared with VKAs, though we could not evaluate the publication bias due to the limited number of enrolled studies. Rivaroxaban was noted with the highest risk of GIB in current analyses.

We set the primary outcome of “any” GIB instead of “major” GIB because clinical decision making is often affected by the GIB signs related to NOACs and VKAs. A major GIB analysis was also performed and the result was similar to any GIB (Appendix A). Except for major GIB, physicians usually hold NOACs or VKAs if the other GIB conditions were suspected, not only focusing on major GIB. Therefore, our study was close to the real-world circumstances. Additionally, we excluded the studies which probably used the identical database or the same population to increase the validity of the meta-analysis. We recruited only Asian patients to emphasize the clinical practicality when physicians prescribe NOACs.

Our study revealed NOACs GI safety versus VKAs and presented different NOACs versus VKAs for the Asian AF population. Previous systematic reviews and meta-analyses reported mainly non-Asians and did not separate AF from the other diseases that also need NOACs treatment. One systematic review and meta-analysis revealed a similar risk of major GIB between NOACs and conventional anticoagulants [55]. However, it included patients with AF and venous thromboembolism (VTE). Among the AF patients, they were almost all non-Asians. The other systematic review and meta-analysis enrolling data from RCTs and real-world studies reported no significant difference in the risk of major GIB between the patients receiving NOACs and conventional anticoagulants. Rivaroxaban users had a 39% increase in the risk for major GIB [56]. However, the recruited patients were nearly from the non-Asia regions. Furthermore, they did not focus on the AF population and not consider the dosage difference between the enrolled studies. Another large-scale network meta-analysis showed that apixaban and edoxaban had the most favorable major GIB safety profile, while rivaroxaban and dabigatran were the least safe [57]. Although the primary outcome was similar to our study and the population was large, it did not focus on Asian AF population and might cause selection bias. We used a more precise statistical method such as HRs, which can represent instantaneous risk over the study period time, or some subset thereof. HRs suffer somewhat less from selection bias concerning the endpoints chosen and can indicate the risks that happen before the endpoint. Therefore, our study could offer a significant and favorable choice of NOACs for clinicians and give patients medical advice about the real GIB risk data.

Different from other areas of the world, NOACs are beneficial for the Asian population and result in less GIB risk. In the non-Asian population, the use of NOACs seems to cause a higher risk of GIB than the Asian population. Holster et al. revealed an increased risk of GIB among NOAC users compared with standard care (pooled OR = 1.45), although significant heterogeneity existed regarding the choice of drugs and the indications of anticoagulation [11]. The other meta-analysis recruiting mainly studies from the USA, New Zealand, and Europe revealed a slightly higher risk of GIB with dabigatran compared with VKAs. In contrast, no significant difference was found between rivaroxaban and VKAs for GIB risk [58]. Another meta-analysis showed that rivaroxaban, high-dose dabigatran, and edoxaban should not be prescribed to patients with high GIB risk [59]. However, this study did not solely enroll Asians.

We disclosed that overall NOACs presented better than VKAs in GIB for the Asian AF population. Previously, a new score system which was named “SAMe-TT2R2” could predict the quality of anticoagulation control among patients with AF on VKAs [60]. Based on Chan’s study, the time in therapeutic range (TTR) decreased progressively with increasing SAMe-TT2R2 score (*p*: 0.016). When the cut-off value of SAMe-TT2R2 score was set at 2, the sensitivity and specificity to predict TTR < 70% were 85.7% and 17.8%, respectively [61]. In the Chinese AF patients, the SAMe-TT2R2 score has a good correlation with TTR. For example, a female Asian’s SAMe-TT2R2 score is at least three, high in the baseline. Then low TTR could cause VKA-related GIB. Therefore, NOACs are a better choice for Asians than VKAs.

We recruited two RCTs in this meta-analysis. Only one enrolled RCT showed some concerns in bias due to deviations from intended interventions because the selected group using NOACs might have less GIB risk. It also showed some concerns in bias in the selection of the reported result. Nevertheless, these two RCTs were very significant for this study because there were few RCTs about NOACs GIB in Asia. Although RCTs have advantages in the GRADE system, they may not reflect the real situations of the AF population in Asia. First, RCTs with limited follow-up can potentially underestimate the long-term benefits of treatment and may fail to detect delayed hazards. A post-trial follow-up of RCTs, which means extended follow-up starting after the end of the scheduled period of the original trial is needed. It is essential not only to define the impact of a long-term intervention but also to ascertain the safety profile. Moreover, potential hazards may not be obvious during the duration of trial follow-up [62]. Second, RCTs usually pay attention to major GIB only, which might lead to an underestimation of all GIB. Third, we need real-world data to distinguish GIB risk from different NOAC because it is impossible to perform head-to-head RCTs currently. In addition, the Asian AF patients only accounted for about 10% in the pivotal RCTs [54]. Therefore, we need to recruit the postmarketing studies and the retrospective observational studies for more accurate data. Besides, physicians might avoid prescribing NOACs for the patients at high risk of GIB in the real-world clinical conditions. Our study contained two RCTs and four retrospective studies, which was close to the real-world practice situations. They were also strictly evaluated by the current appraisal tools from the Cochrane system. There were still some biases in our enrolled retrospective studies. They had serious performance in bias in the selection of participants into the study, which was probably due to ICD codes not precise in the diagnosis from the database. There was a low risk of bias in the classification of interventions and bias in missing data because the clinical setting was prominent while GIB happened and the database was intact in several Asian countries/regions (Taiwan, Korea, Japan, China/Hong Kong, etc.). A serious risk of bias due to deviations from intended interventions was originated from unseen biases such as the methods of study design, patient’s lifestyle and eating habits, body mass index, and alcohol/betel nut/smoking. A serious risk of bias in the measurement of outcomes happened because sometimes clinical conditions such as bloody sputum or food digestion color were mistaken as GIB. A moderate risk of bias in the selection of the reported result was evaluated because there might be some negative result data not reported. The definite conclusions could not be just based on these studies.

Recently, the FDA issued a renewed dabigatran safety announcement, which reported a higher GIB risk (HR = 1.28; 95% CI, 1.14 to 1.44) in contrast with warfarin [63,64]. Besides, the postmarketing pharmacovigilance studies showed adverse drug reactions of GIB in Japan, Australia, Canada, and the USA [41,65,66,67,68].

NOACs, mainly rivaroxaban and dabigatran, were considered more dangerous in GIB. However, the apixaban and the edoxaban observational studies re-defined the GIB risk. One study revealed the non-major bleeding (including GIB) was substantially less frequent in apixaban than in warfarin [69]. Another first head-to-head Korean study made a comparison of the effectiveness and safety between rivaroxaban and edoxaban and showed that edoxaban had a trend toward less GIB [70]. The results from these two studies were similar to our study. However, we still need more observational studies from other countries in Asia to establish the NOACs GI safety profile in the future.

Our studies had some limitations. First, not all Asian countries were included, and the results could not be applied to the whole Asian population. Second, the apixaban and the edoxaban head-to-head studies are still lacking because the marketed time was shorter than that of dabigatran and rivaroxaban. Third, we calculated the HR of GIB from different NOACs compared with VKAs, but our enrolled studies did not uniformly use the same definition of GIB event and did not describe the source of GIB at all. Finally, we did not focus on the meta-analyses of the different doses of NOACs versus VKAs for GIB risk because few studies included this concern. In our enrolled articles, only Yamashita et al. [49] and Chan et al. [51] had analyzed the GIB risk of standard-dose and low-dose NOACs compared with VKAs. The result of meta-analyses was shown in Appendix A, which suggested that low-dose NOACs was significantly associated with a lower risk of GIB than standard-dose NOACs compared with VKAs.

## 5. Conclusions

This meta-analysis revealed that NOACs could cause less GIB risk than VKAs. Among the NOACs compared with VKAs, apixaban was associated with the least risk of GIB. We need further comparative studies of different NOACs to confirm which NOAC has the best GI safety for the Asian AF patients and to determine the best dosage regimen of different NOACs.

## Figures and Tables

**Figure 1 ijerph-18-00137-f001:**
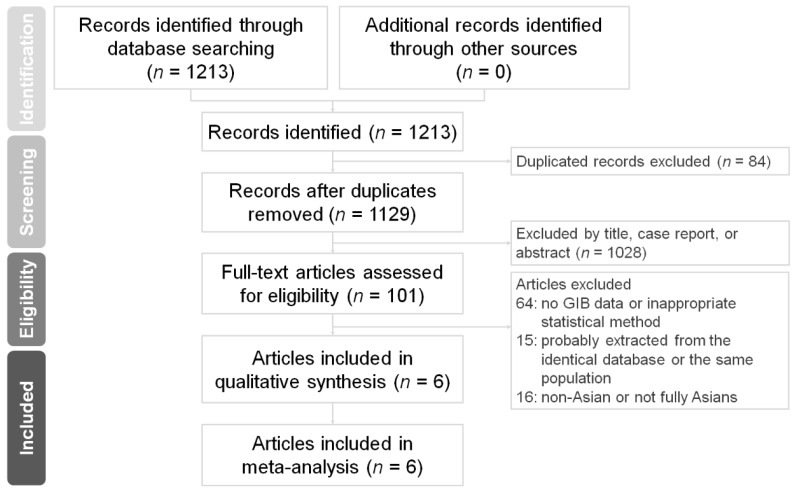
Preferred Reporting Items for Systematic Reviews and Meta-Analyses (PRISMA) flow chart. Preferred reporting items for systematic reviews and meta-analyses flow diagram for the searching and identification of included studies.

**Figure 2 ijerph-18-00137-f002:**
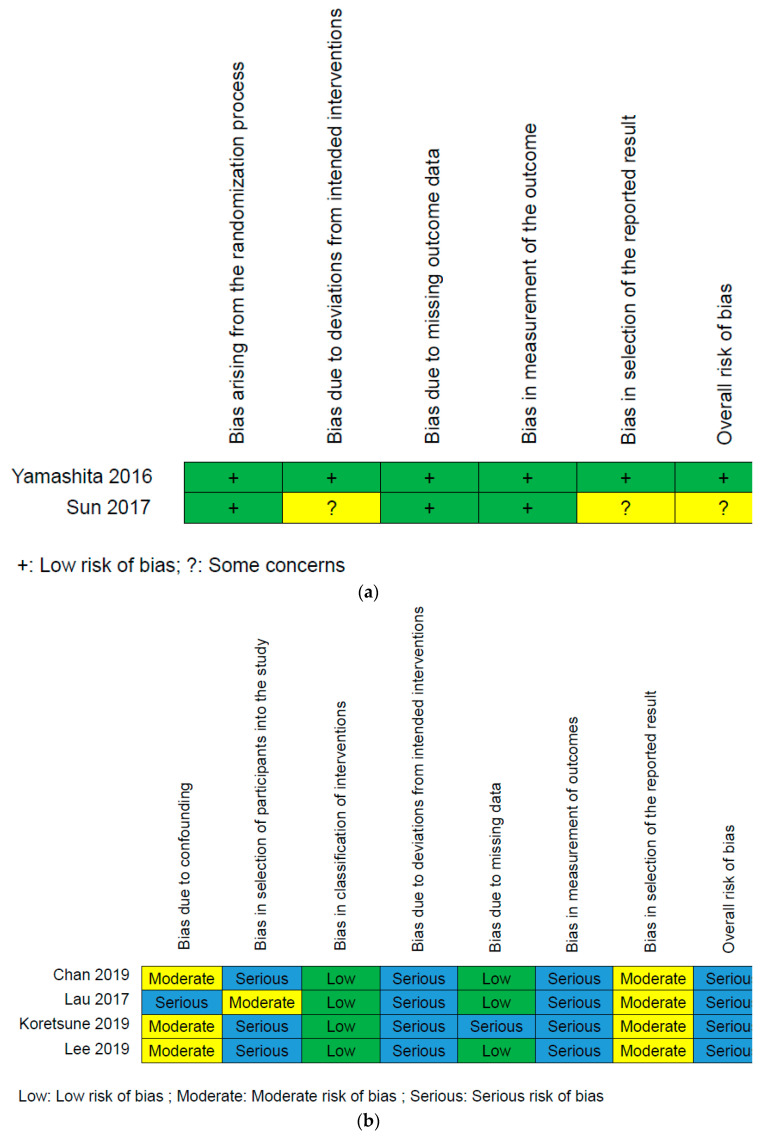
(**a**) Summary of each randomized controlled trial (RCT) appraised by ROB 2.0. (**b**) Summary of each retrospective study evaluated by ROBINS-I.

**Figure 3 ijerph-18-00137-f003:**
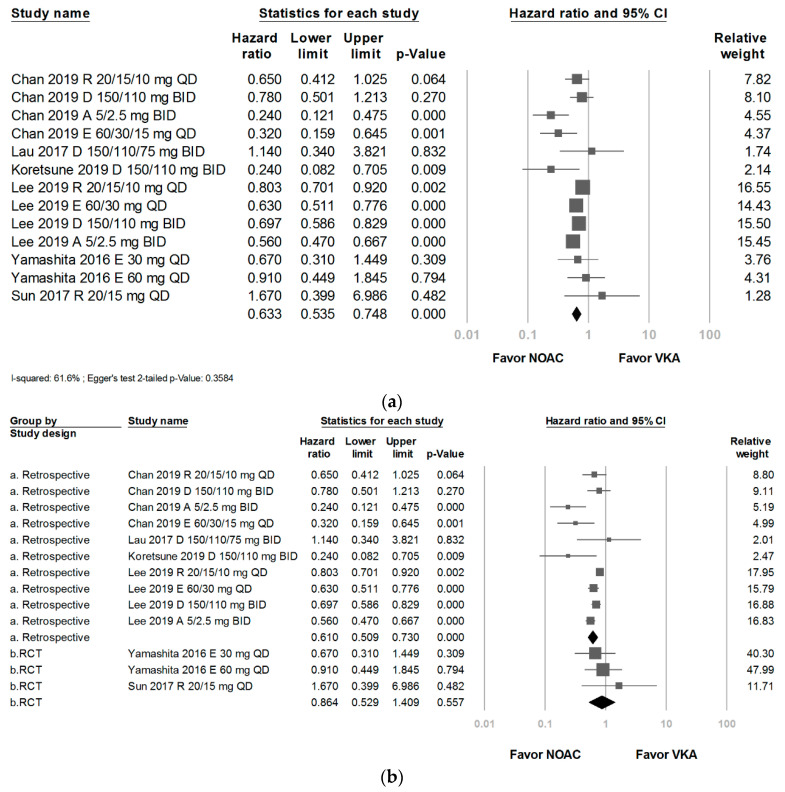
(**a**) Forrest plot of comparison: all novel oral anticoagulants versus vitamin K antagonists. (**b**) Forrest plot of comparison: the retrospective studies subgroup and the RCTs subgroup.

**Figure 4 ijerph-18-00137-f004:**
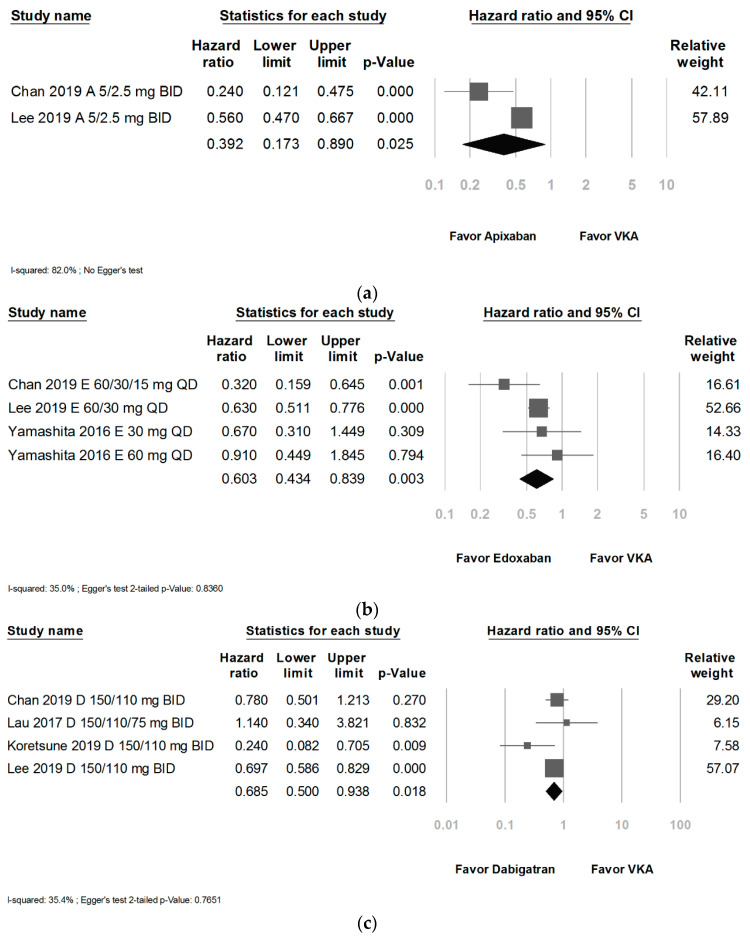
(**a**) Forrest plot of comparison: the subgroup of apixaban versus vitamin K antagonists. (**b**) Forrest plot of comparison: the subgroup of edoxaban versus vitamin K antagonists. (**c**) Forrest plot of comparison: the subgroup of dabigatran versus vitamin K antagonists. (**d**) Forrest plot of comparison: the subgroup of rivaroxaban versus vitamin K antagonists.

**Figure 5 ijerph-18-00137-f005:**
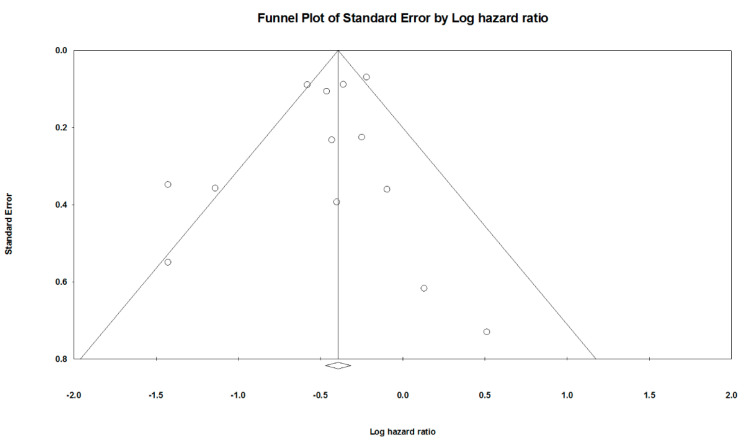
Funnel plots of pooled randomized controlled trials and retrospective studies.

**Table 1 ijerph-18-00137-t001:** Summary of the retrieved articles for gastrointestinal bleeding risk of non-vitamin K antagonist oral anticoagulants and vitamin K antagonists in the Asian atrial fibrillation patients.

Author, Year	Intervention	Design	Number of Patients (NOAC/VKA)	Average Age	Data Source	Country/Region	Outcome
Chan, 2019 [51]	R: 20/15/10 mg QD D: 150/110 mg BID A: 5/2.5 mg BID E: 60/30/15 mg QD	Retro	69922/19761	R: 75; D: 75 A: 75; E: 75 VKA: 75	Taiwan National Health Insurance Database	Taiwan	Favor apixaban and edoxaban for less GIB and a trend toward rivaroxaban and dabigatran for less GIB than VKAs
Lau, 2017 [52]	D: 150/110/75 mg BID	Retro	2580/2580	D: 74 VKA: 74	CDARS * of the HongKong Hospital Authority	China/Hong Kong	A trend toward VKAs for less GIB than dabigatran
Korestune, 2019 [53]	D: 150/110 mg BID	Retro	4606/4606	D: 74 VKA: 73	Medical Data Vision (MDV, Tokyo, Japan)	Japan	Favor dabigatran for less GIB than VKAs
Lee, 2019 [54]	R: 20/15/10 mg QD D: 150/110 mg BID A: 5/2.5 mg BID E: 60/30 mg QD	Retro	91383/25420	R: 71; D: 71 A: 71; E: 71 VKA: 71	Korean Health Insurance Review service	Korea	Favor overall NOACs for less GIB than VKAs
Yamashita, 2016 [49]	E: 60/30 mg QD	RCT	1221/592	East Asian: 70	ENGAGE AF-TIMI 48 ^†^ subanalysis	China, Japan, Taiwan, South Korea	A trend toward edoxaban for less GIB than VKAs
Sun, 2017 [50]	R: 20/15 mg QD	RCT	249/246	China: 71	ROCKET AF trial ^#^	China	A trend toward VKAs for less GIB than rivaroxaban

Abbreviations: R, Rivaroxaban; D, Dabigatran; A, Apixaban; E, Edoxaban; VKAs, vitamin K antagonists; Retro: Retrospective studies; RCT: randomized controlled trial. *: CDARS represents Clinical Data Analysis and Reporting System. ^†^: ENGAGE AF-TIMI 48 represents Effective Anticoagulation with Factor Xa Next Generation in Atrial Fibrillation-Thrombolysis in Myocardial Infarction 48. ^#^: ROCKET AF represents Rivaroxaban Once-daily, Oral, direct Factor Xa Inhibition Compared with Vitamin K Antagonism for Prevention of Stroke and Embolism Trial in Atrial Fibrillation.

**Table 2 ijerph-18-00137-t002:** Meta-analyses of gastrointestinal bleeding in different groups.

Group	HR [95% CI]	I^2^%	Egger’s Test *p*-Value	Quality of the Evidence (GRADE)
Overall	0.63 [0.54 to 0.75]	61.6	0.3584	Very low (a, b, g)
RCT subgroup	0.86 [0.53 to 1.41]	0	0.4281	High (a, c, g, h)
Retrospective subgroup	0.61 [0.51 to 0.73]	68.9	0.1189	Very low (a, b, g)
Apixaban subgroup	0.39 [0.17 to 0.89]	82.0	NA	Very low (a, b, e, f, g)
Edoxaban subgroups	0.60 [0.43 to 0.84]	35.0	0.8360	Low (a, g)
Dabigatran subgroup	0.69 [0.50 to 0.94]	35.4	0.7651	Low (a, g)
Rivaroxaban subgroup	0.79 [0.70 to 0.90]	0	0.8630	Low (a, g)

Quality of the evidence: High certainty: We are very confident that the true effect lies close to that of the estimate of the effect. Moderate certainty: We are moderately confident in the effect estimate: The true effect is likely to be close to the estimate of the effect, but there is a possibility that it is substantially different. Low certainty: Our confidence in the effect estimate is limited: The true effect may be substantially different from the estimate of the effect. Very low certainty: We have very little confidence in the effect estimate: The true effect is likely to be substantially different from the estimate of effect. a Downgraded due to risk of bias. b Downgraded due to inconsistency. c Downgraded due to imprecision. d Downgraded due to indirectness. e Downgraded due to publication bias. f Upgraded due to large effect. g Upgraded due to dose-response. h Upgraded due to plausible confounders.

**Table 3 ijerph-18-00137-t003:** Results of meta-regressions for probable covariates.

Covariate	Coefficient	Standard Error	2-Sided *p*-Value	Tau Squared	Tau	Q	df
Age	−0.0789	0.0457	0.0842	0.0353	0.1880	27.39	11
Female ratio	0.1498	1.8083	0.9340	0.0425	0.2061	31.22	11
Publication year	−0.1309	0.0985	0.1835	0.0397	0.1994	30.05	11

## Data Availability

Data available in a publicly accessible repository.

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
