# Peer review of "The Risk of Gastrointestinal Bleeding between Non-Vitamin K Antagonist Oral Anticoagulants and Vitamin K Antagonists in the Asian Atrial Fibrillation Patients: A Meta-Analysis"

_ijerph, 2020, doi:10.3390/ijerph18010137_

Round 1

Reviewer 1 Report

There are few concerns.

1. The authors did not highlight in the discussion that there was serious bias in the retrospective studies. It would be inappropriate to make definite conclusions just based on these studies. 

2. Why does authors feel Asian patients should have a different gi bleeding risk. Can you bring up some hypothesis.

3. Why was a sub/separate analysis of major bleeding rather than any bleeding not performed? It would be very useful to see what trends we obtain for major bleeding.  

Minor corrections: grammer error : There is a need for editing and english language style correction needed in addition to the ones mentioned below. 

Line 35: prevalence was; 

Line 37: higher probability, not higher the probability.

Line 38: grammatical error.

Line 40: change the sentence.

Line 41: wrong grammer, please correct

Line 95 – need grammer correction

Line 14 – correction needed.

Lilne 134 : why probably?

Figure – articles excluded à keep the numbers in same format as the rest of the the figure (n=  ) .

Table 1 – table needs to be made symmetrical and arranged in accordance with the heading

Line 169 grammer is not correct.

Line 177 – change

Line 181 – correct grammer.

Line 285 – grammer error.

Line 297, line 327 grammer correction. Line 339 --

Comment on why Asian population there is difference in risk of gi bleed compared to tohers.

Line 339 – why would you underestimate proportion of adverse effects in randomized controlled study. It just depends on the follow up duration.

Author Response

Dear editor and reviewer:

Thank you for your kindly and detailed review. We’ve provided a point-by-point revision or response to all of your comments in the following table. In the revised manuscript, all the changes are highlighted in red color. An identical manuscript without highlights has been submitted as well. We deeply apologize for the delayed reply and appreciate your valuable opinions which make this article more integrated. Thank you.

There are few concerns.

-Thank you for considering our manuscript for further revisions.

(Pages and lines were in accordance with the revised manuscript without highlights.)

1.The authors did not highlight in the discussion that there was serious bias in the retrospective studies. It would be inappropriate to make definite conclusions just based on these studies. 

-Yes, we agreed with the advice of the reviewer. We had added this part in our manuscript (page 13, line 340-351).

2.Why does authors feel Asian patients should have a different gi bleeding risk. Can you bring up some hypothesis.

-Yes, we have added some hypotheses in our introduction parts. Please check the revised manuscript (page 2, line 55-58).

3.Why was a sub/separate analysis of major bleeding rather than any bleeding not performed? It would be very useful to see what trends we obtain for major bleeding. 

-Yes, we appreciate the advice of the reviewer. We had added the meta-analysis of major GI bleeding in our supplementary figure 4 (page 12, line 275-277). The outcome was similar to any GI bleeding.

Minor corrections: grammer error : There is a need for editing and english language style correction needed in addition to the ones mentioned below. 

Line 35: prevalence was; 

-Yes, we have corrected the sentence (page 1, line 36)

Line 37: higher probability, not higher the probability.

-Yes, we have corrected the sentence (page 1, line 38)

Line 38: grammatical error.

-Yes, we have corrected the sentence (page 1, line 38-39)

Line 40: change the sentence.

-Yes, we have corrected the sentence (page 1, line 41-44)

Line 41: wrong grammer, please correct

-Yes, we have corrected the sentence (page 1, line 44)

Line 95 – need grammer correction

-Yes, we have corrected the sentence (page 3, line 100-101)

Line 14 – correction needed. (Does line 14 mean line 114?)

-Yes, we have corrected the sentence (page 3, line 117-119)

Line 134 : why probably?

-Because some studies were published from the same database or the same population from a country or region, we chose the most representative study to perform further analyses.

Figure – articles excluded à keep the numbers in same format as the rest of the the figure (n=  ) .

-Yes, we have corrected the Figure 1.

Table 1 – table needs to be made symmetrical and arranged in accordance with the heading

-Yes, we have made symmetrical and arranged in accordance with the heading in Table 1.

Line 169 grammer is not correct.

-Yes, we have corrected the sentence (page 7, line 163).

Line 177 – change

-Yes, we have corrected the sentence (page 7, line 167) and transfer it to discussion part (page 13, line 340-351).

Line 181 – correct grammer.

-Yes, we have corrected the sentence (page 7, line 169).

Line 285 – grammer error.

-Yes, we have corrected the sentence (page 11, line 269).

Line 297, line 327 grammer correction. Line 339 --

-Yes, we have corrected the sentence (page 12, line 283-284; page 12, line 313-314; page 13, line 329-330).

Comment on why Asian population there is difference in risk of gi bleed compared to others.

-Yes, we have described the difference on page 2, line 44-47. Asian AF patients under vitamin K antagonists (VKAs, warfarin) use easily encountered bleeding events and would seldom reach optimal international normalized ratio control when taking VKAs.

The other explanation is on page 12, line 313-321. There are many factors affecting drug effects and metabolism between the Asian patients and the non-Asian patients. The SAMe-TT2R2 score is a good tool for clinicians to make decisions in anticoagulation use.

Line 339 – why would you underestimate proportion of adverse effects in randomized controlled study. It just depends on the follow up duration.

-Yes, we agree with the concept of the reviewer. We had modified our sentences (page 12-13, line 322-329) according to the previous reference (Llewellyn-Bennett, R. et al. 2016).

Reviewer 2 Report

In the current manuscript, the authors performed a meta-analysis with several up-to-date trials and retrospective studies. They demonstrated that non-vitamin K antagonist oral anticoagulants (NOACs) are safer than vitamin K antagonists (VKAs) when administrated to prevent atrial fibrillation (AF) patients from thromboembolic events, in terms of the gastrointestinal bleeding (GIB) risk. They further compared the GIB risks of four different NOACs. Their results recapitulate previous reports about the NOACs and VKAs safety study in AF patients, which can be improved by addressing the following comments.

  1. The manuscript is properly written in English, but there are several claims exaggerating the novelty of this study. For example, in Line 50-54, the authors claimed that "There is no specific systematic review and meta-analysis for NOACs versus VKAs related to GIB risk in Asia. Furthermore, the optimal NOAC in Asian AF patients in terms of GIB risk was not clear". However, in the Wang et al. 2015 paper, the authors clearly made the conclusion that standard-dose NOACs are safer than VKAs about GIB events in Asians with convincing results, although it is not the major point made in that paper. Please be fair when making claims and referring to others' publications.
  2. The conclusion that "apixaban is the most favorable NOAC" is not convincingly supported by their results. When analyzing the efficacy of four different NOACs, they only made the comparison between each NOAC and VKA, but not between different NOACs, which is necessary if they want to claim one is better than the other. Additionally, the reviewer observed that there were only two studies containing apixaban statistics, which are less than the other three NOACs and may cause biased conclusion.
  3. The authors failed to make it clear what is the definition of GIB event, how they make sure that the GIB event is characterized uniformly across different studies.
  4. The authors merged different doses of NOACs from one study as one single sample statistics. It is necessary to elucidate whether there are dose-dependent effects in Asians about GIB events, especially it has been shown that low-dose and standard-dose NOACs have distinct influence on different races (Wang et al. 2015).
  5. As well-educated scientists, it is hard to understand why the authors are not aware of the fact that Hong Kong is not an independent country. In many sections of the manuscript, the authors seem to indicate Hong Kong as a country on purpose. Everyone can have their own opinions and should be respected. But it is inappropriate and unprofessional to provoke a political dispute in a scientific paper, which may not only hurt the feeling of some other groups of scientists but also put the reputation of the journal at risk. Please change the terminology Country to Country/Region.

Author Response

Dear editor and reviewer:

Thank you for your kindly and detailed review. We’ve provided a point-by-point revision or response to all of your comments in the following table. In the revised manuscript, all the changes are highlighted in red color. An identical manuscript without highlights has been submitted as well. We deeply apologize for the delayed reply and appreciate your valuable opinions which make this article more integrated. Thank you.

In the current manuscript, the authors performed a meta-analysis with several up-to-date trials and retrospective studies. They demonstrated that non-vitamin K antagonist oral anticoagulants (NOACs) are safer than vitamin K antagonists (VKAs) when administrated to prevent atrial fibrillation (AF) patients from thromboembolic events, in terms of the gastrointestinal bleeding (GIB) risk. They further compared the GIB risks of four different NOACs. Their results recapitulate previous reports about the NOACs and VKAs safety study in AF patients, which can be improved by addressing the following comments.

-Thanks a lot! We had performed revisions under the suggestions.

(Pages and lines were in accordance with the revised manuscript without highlights.)

1.The manuscript is properly written in English, but there are several claims exaggerating the novelty of this study. For example, in Line 50-54, the authors claimed that "There is no specific systematic review and meta-analysis for NOACs versus VKAs related to GIB risk in Asia. Furthermore, the optimal NOAC in Asian AF patients in terms of GIB risk was not clear". However, in the Wang et al. 2015 paper, the authors clearly made the conclusion that standard-dose NOACs are safer than VKAs about GIB events in Asians with convincing results, although it is not the major point made in that paper. Please be fair when making claims and referring to others' publications.

-Yes, we have modified our sentences to avoid inappropriate claims, and we refer to others’ publications. Please check the revised manuscript (page 2, line 54-61).

2.The conclusion that "apixaban is the most favorable NOAC" is not convincingly supported by their results. When analyzing the efficacy of four different NOACs, they only made the comparison between each NOAC and VKA, but not between different NOACs, which is necessary if they want to claim one is better than the other. Additionally, the reviewer observed that there were only two studies containing apixaban statistics, which are less than the other three NOACs and may cause biased conclusion.

-Yes, we agree with the comment of the reviewer. We have modified our sentences to an appropriate conclusion. Please check them in our revised manuscript (page 1, line 28-30; page 14, line 377-379).

3.The authors failed to make it clear what is the definition of GIB event, how they make sure that the GIB event is characterized uniformly across different studies.

-Yes, we agreed with the comment of the reviewer. However, due to different study designs, our enrolled studies did not use the uniform definition of GIB event. Though there could be some biases, we could find the trend of the GIB risk using NOACs versus VKAs in our meta-analysis according to current trials and real-world studies, including a large number of patients. The statistical power of big case numbers might help offset differences between studies. We still described the above limitation to balance the report. Please check it in our revised manuscript (page 13, line 367-369).

4.The authors merged different doses of NOACs from one study as one single sample statistics. It is necessary to elucidate whether there are dose-dependent effects in Asians about GIB events, especially it has been shown that low-dose and standard-dose NOACs have distinct influence on different races (Wang et al. 2015).

-Yes, we agreed with the advice of the reviewer. However, currently in our enrolled articles, only two articles (Chan et al. 2019 and Yamashita et al. 2016) demonstrated that compared with VKAs, standard-dose NOACs had a higher risk of GIB than low-dose NOACs. We have added this sub-analysis in our discussion, and the result revealed that low-dose NOACs significantly lowered the GIB risk than stand-dose NOACs versus VKAs. Please check the revised manuscript (page 13, line 369-374) and the supplementary figure 5.

5.As well-educated scientists, it is hard to understand why the authors are not aware of the fact that Hong Kong is not an independent country. In many sections of the manuscript, the authors seem to indicate Hong Kong as a country on purpose. Everyone can have their own opinions and should be respected. But it is inappropriate and unprofessional to provoke a political dispute in a scientific paper, which may not only hurt the feeling of some other groups of scientists but also put the reputation of the journal at risk. Please change the terminology Country to Country/Region.

-Yes, we agree with the reviewer's advice that the inappropriate terminology of country/region might cause a political dispute in a scientific paper. We have already changed the terminology “Hong Kong” to “China/Hong Kong” in our revised manuscript (Table 1; page 13, line 345).

Round 2

Reviewer 2 Report

Thank you for all your point-to-point revisions. I believe that your work will contribute to the field of study.